# The Use of Fast-Acting Insulin Topical Solution on Skin to Promote Surgical Wound Healing in Cats

**DOI:** 10.3390/ani14091358

**Published:** 2024-04-30

**Authors:** L. Miguel Carreira, Rúben Silva, João Alves, Filipa Inácio, Graça Pires, Pedro Azevedo

**Affiliations:** 1Anjos of Assis Veterinary Medicine Centre—CMVAA, Rua D.ª Francisca da Azambuja Nº9-9A, 2830-077 Barreiro, Portugal; rvdsilva17@gmail.com (R.S.); pedro.almeida.azevedo@gmail.com (P.A.); 2Faculty of Veterinary Medicine, University of Lisbon (FMV/ULisboa), Av. da Universidade Técnica, 1300-477 Lisbon, Portugal; finacio@fmv.ulisboa.pt (F.I.); gpires@fmv.ulisboa.pt (G.P.); 3Interdisciplinary Centre for Research in Animal Health (CIISA), University of Lisbon (FMV/ULisboa), Av. da Universidade Técnica, 1300-477 Lisbon, Portugal; 4Associate Laboratory for Animal and Veterinary Sciences (AL4AnimalS), 1300-477 Lisbon, Portugal; 5Faculty of American Laser Study Club—ALSC, Altamonte Springs, FL 32714, USA; 6Divisão de Medicina Veterinária, Guarda Nacional Republicana (GNR), Rua Presidente Arriaga, 9, 1200-771 Lisbon, Portugal; alves.jca@gnr.pt

**Keywords:** cat, insulin, healing, wound, surgery

## Abstract

**Simple Summary:**

A study investigated the effects of applying fast-acting insulin directly to surgical wounds in cats. The aim was to see if insulin could speed up the healing process. Sixty healthy female cats undergoing spaying were used in the study. Each surgical site was split into two sections: one receiving insulin treatment and the other serving as a control. The results showed that wounds treated with insulin healed faster, with fewer complications such as liquid buildup and stitch removal. The insulin promoted early infiltration of healing cells, leading to quicker wound closure. This study suggests that applying insulin directly to surgical wounds in cats can improve healing and represents a novel approach in veterinary medicine.

**Abstract:**

Wound healing is a complex biological process involving a coordinated sequence of events aimed at restoring tissue integrity and function. Recent advancements in wound care have introduced novel therapies, with topical insulin application emerging as a promising strategy for promoting tissue healing. This study, involving 60 female cats (*n* = 60) undergoing elective spaying, aimed to evaluate the effects of topical fast-acting insulin on the healing process of surgical wounds. Each surgical suture was divided into two regions: the control zone (Zcr) without insulin application and the study zone (Zst), where insulin was applied topically for 10 min every 24 h over eight consecutive days. Assessment of suture healing was conducted using an adapted scale at two time points post-surgery: T1 (day 2) and T2 (day 8). Statistically significant differences were registered in the final healing scale scores between Zcr and Zst (*p* < 0.022), as well as for the parameter of regional fluid (*p*-value = 0.017). Additionally, at T2, all Zst regions exhibited wound closure, whereas Zcr did not, although not in a statistically significant manner. The observed discrepancy at T2 between the Zcr and Zst regions may suggest a potential benefit of utilizing insulin. No side effects resulting from the insulin topical application performed by the tutors were recorded in the Zst suture group. This study represents the first exploration of the benefits of topical insulin application for surgical wound healing in cats.

## 1. Introduction

In recent years, the topical application of growth factors to enhance wound healing has gained popularity. However, challenges arise due to the often low concentrations of these factors in the wound bed, primarily attributed to decreased supply or natural degradation [1]. While some dressing systems may mitigate this issue, a critical evaluation of alternative therapeutic approaches is warranted. Insulin, a growth factor with multiple physiological functions, including its primary role in reducing blood glucose levels, has garnered attention as a promising candidate for wound healing research, both in animal models and human clinical practice, due to its distinct mechanisms of action emerging evidence of its involvement in various cellular processes crucial for wound healing [1,2]. Unlike growth factors, insulin’s mode of action is not solely dependent on its concentration in the wound bed. Beyond its canonical function in glucose uptake and metabolism, insulin exerts pleiotropic effects on diverse cell types implicated in tissue repair. These effects, through various pathways, include the stimulation of cell proliferation, collagen synthesis, and angiogenesis, thereby promoting wound closure and tissue regeneration. Moreover, insulin’s ability to modulate inflammation, oxidative stress, and matrix remodeling further underscores its therapeutic potential in wound healing [2]. In 1954, Lawrence et al. [3] studied the effects of insulin in hypophysectomized rats, proposing that insulin would be responsible for nitrogen retention, amino acid uptake, protein synthesis, and the inhibition of protein catabolism, all crucial factors in promoting wound tissue healing. In diabetic individuals, wound healing is impaired. Thalhimer and Achar et al. [4] demonstrated that these individuals exhibit reduced expression of a factor called IGF-1, which is only expressed in fibroblast and keratinocyte cells of the dermis and epidermis [4,5,6]. IGF-1, a protein with a sequence very similar to insulin, has been shown to promote granulation tissue synthesis in in vivo studies. It is a potent stimulator of mitogenesis, promoting cell survival in wounds in an autocrine, paracrine, and endocrine manner. IGF-1 levels typically increase at the wound site approximately 1 to 3 days after its appearance. In 2012, Abo El Asrar et al. [7] demonstrated a relationship between IGF-1 levels and inflammation markers, namely IL-6 and IL-8, in diabetic patients, highlighting decreased IGF-1 levels and increased inflammation markers in such patients. Fibroblasts and keratinocytes in the skin possess insulin receptors (IR) with varying binding affinities [8]. The presence of IR in these cells, coupled with insulin’s ability to activate phosphatidylinositol kinase-3 transduction pathways and mitogenic proteins with kinase activity, supports the efficacy of topical insulin application in promoting wound healing. Of particular interest is insulin’s interaction with insulin-like growth factor 1 (IGF-1) signaling pathways, which play a pivotal role in mediating its effects on cell proliferation and differentiation. By amplifying the activity of endogenous growth factors like IGF-1, insulin potentiates their pro-healing effects, offering a synergistic approach to enhance tissue repair. Additionally, insulin’s systemic effects extend beyond its local application, influencing systemic factors such as glucose homeostasis and vascular function and enhancing tissue perfusion, which is integral to the overall wound healing process. This systemic modulation complements its local actions, creating a favorable physiological environment conducive to optimal wound repair. In summary, while growth factors may face challenges related to their concentration in the wound bed, insulin’s multifaceted mechanisms of action and systemic effects make it a viable candidate for promoting wound healing, capable of overcoming some of the limiting factors described in the sentence. In rodent models, topical insulin has been shown to accelerate wound closure, enhance granulation tissue formation, and improve tensile strength in healed wounds [9,10,11,12,13,14]. Moreover, insulin exhibits anti-inflammatory properties, reducing pro-inflammatory cytokine levels and modulating immune cell function within the wound microenvironment [15,16]. Translating these findings to human clinical practice holds significant promise for improving wound care outcomes. Clinical studies have reported favorable outcomes with topical insulin therapy in various wound types, including diabetic ulcers, pressure ulcers, and surgical wounds [17,18]. By harnessing insulin’s pleiotropic effects, clinicians can potentially expedite wound healing, reduce the risk of complications, and improve patient outcomes. Furthermore, topical insulin therapy offers several advantages over traditional wound care approaches. Unlike systemic insulin administration, which may be associated with hypoglycemia risk, topical application allows for localized delivery of insulin to the wound site, minimizing systemic side effects [19]. Additionally, insulin is readily available, cost-effective, and well-tolerated, making it a practical option for widespread clinical use [20]. In the present study, conducted in cats, aimed to evaluate the outcomes of using fast-acting insulin topically in the healing process of surgical wounds.

## 2. Materials and Methods

The study comprised a sample of 60 (*n* = 60) female cats, all of whom were inpatients undergoing elective ovariohysterectomy surgery. Approval for the study was obtained from the Animal Ethics and Welfare Council (CEBEA) of FMV-ULisboa under code 003/2017. The participation of the animals commenced only after their tutors had signed informed consent forms. All patients received the same medical, analgesic, and anesthetic protocol, including amoxicillin + clavulanic acid (10 mg/kg; subcutaneous, Synulox^®^, Portugal), buprenorphine (0.02 mg/kg; intramuscular, Bupaq^®^, Portugal), and tolfenamic acid (4 mg/kg; subcutaneous, Tolfedine^®^, Poland). Anesthetic induction was achieved using a combination of ketamine (5–7.5 mg/kg; intramuscular, Nimatek^®^, The Netherlands) and dexmedetomidine (40 μg/kg; intramuscular, Sedadex^®^, The Netherlands), followed by maintenance with isoflurane (Isofluo^®^, Portugal). Dexmedetomidine was reversed shortly after surgery using atipamezole (0.1 mg/kg; intramuscular, Revertor^®^, Germany). To minimize bias, the surgical procedure was standardized and performed by the same surgeon. Each surgical suture was divided into two regions: the cranial region served as the control zone (Zcr), while the caudal region served as the study zone (Zst), where a saline solution with fast-acting insulin was topically applied. Each animal tutor received a kit containing eight compresses (5 cm × 5 cm) and an 8 mL syringe with a fast-acting insulin solution (Actrapid^®^) in a concentration of 2 IU/mL 0.9% NaCl solution. Tutors were instructed by the researcher on how to apply the solution uniformly over the Zst. The solution was applied topically to the skin only at Zst using a compress soaked in 1 mL of the solution for a period of 10 min, every 24 h for eight consecutive days. The tutors were also asked to photograph the suture at the two time points considered: T1 (day 2) and T2 (day 8). For the evaluation of the suture healing process, an adaptation of the post-surgical healing evolution scale developed by Vítor and Carreira [21] was used, considering four parameters: (1) skin color, (2) hematoma (presence or absence), (3) regional fluid (presence or absence), and (4) wound closure. Each parameter was scored on an ordinal scale, and the final value was the sum of all scores. The scale considers that for each parameter, a value is selected that is higher the greater the level of tissue changes presented, and lower final scores indicate greater healing rates (Table 1). The data obtained were analyzed using the R^®^ program, version 3.4.0, with the R commander extension. For ordinal qualitative variables, means were calculated, and their distributions were characterized. Normality was assessed using the Shapiro-Wilk test. Measures of dispersion included mean and standard deviation for normally distributed data and median and interquartile range for non-normally distributed data. The non-parametric Kruskal-Wallis test was used for qualitative ordinal variables, with statistical significance set at *p* < 0.05.

## 3. Results

The sample’s characteristics for age, body weight, and healing scale parameters are presented in Table 2. Using the Shapiro-Wilk test, it was concluded that, except for age (*p* = 0.015), none of the others exhibited a normal distribution (*p* > 0.05). Suture healing evaluation considered two different postoperative time points (T1 and T2). At T1, no statistically significant differences were observed between the Zcr and the Zst of individuals for all parameters, contrary to what was registered at T2 (Figure 1, Figure 2, Figure 3 and Figure 4). Utilizing the Kruskal-Wallis test of independent measures, it was verified that at T1, no statistically significant differences in the healing scale final score between Zcr and Zst were observed (*p* = 0.725). The Tukey HSD test confirmed that no statistically significant differences were registered at this time point between the Zcr and Zst regions regarding each parameter, namely skin color (*p* = 0.261), hematoma (*p* = 0.071), regional fluid (*p* = 0.104) and wound closure (*p* = 0.368) (Table 3). The Kruskal-Wallis test of independent measures was also made at T2, showing statistically significant differences in the healing scale final score between Zcr and Zst (*p* < 0.022). The Tukey HSD test confirmed that statistically significant differences were registered only for the parameter regional fluid (*p*-value = 0.017) but not for the others, skin color (*p* = 0.159), hematoma (*p* = 0.323) and wound closure (*p* = 0.465) (Table 3). At T2, all the Zst exhibited wound closure, whereas Zcr did not, although not in a statistically significant manner (Figure 4 and Figure 5). The discrepancy observed at T2 between the Zcr and Zst regions may suggest a potential benefit of utilizing insulin. No side effects were recorded in the suture study regions (Zcr and Zst) resulting from the insulin topical application performed by the tutors, who had been previously instructed on how to apply it over the suture line.

## 4. Discussion

Wound healing is a complex biological process characterized by a synchronized sequence of events aimed at restoring tissue integrity and function. Progressing through four well-defined stages: inflammation, debridement, repair (proliferation), and remodeling or maturation [20,21,22,23,24,25], this intricate process entails the orchestrated interaction of various components, including blood cells, extracellular matrix constituents, parenchymal cells, and soluble mediators [25]. While wound healing conventionally follows a temporal sequence in uncomplicated cases, it is noteworthy that different regions of the wound may manifest varying stages of repair, often overlapping temporally [26]. Recent advancements in wound care have precipitated the emergence of novel therapeutic modalities targeting both acute and chronic wound healing, with the overarching objective of achieving complete lesion closure and restoration of optimal tissue function [27,28]. Notably, among these innovative approaches lies the topical application of insulin, which has exhibited considerable efficacy in promoting tissue healing [29]. In both human and veterinary medicine, the management of fresh surgical wounds mandates meticulous consideration of potential risks, including the potential for toxicity arising from various agents. In human surgery, the wound microenvironment’s integrity can be influenced by factors such as local tissue perfusion, the presence of infection, and the nature of therapeutic interventions employed. Similar considerations apply in veterinary practice, albeit with species-specific variations in wound healing dynamics and pharmacokinetics. Studies in both disciplines emphasize the importance of assessing the potential for toxicity associated with the introduction of substances into a surgical wound. Factors such as the concentration of the agent, its pharmacological properties, and the individual patient’s physiological status must all be carefully considered to mitigate the risk of adverse effects [30,31,32,33]. Additionally, the anatomical site of the surgical wound can significantly influence the likelihood of toxicity. Core anatomical locations may exhibit heightened sensitivity to exogenous substances, necessitating enhanced vigilance in monitoring and management [34,35,36,37]. Insulin, a peptide hormone, is traditionally administered via subcutaneous injection to achieve systemic effects, particularly for glycemic control in diabetic patients. However, it has been used for topical application over wounds, and its transcutaneous absorption, particularly in the context of freshly surgical wounds, can pose potential risks that warrant careful consideration. Under specific conditions, such as when the skin barrier function is compromised or when there’s increased vascular permeability due to surgical trauma, insulin may be absorbed through the skin in an uncontrolled manner, disrupting the intricate processes of wound healing. The epidermis, comprising the outermost layer of the skin, serves as the primary barrier to drug absorption. Insulin molecules can traverse the epidermal layers through passive diffusion or facilitated transport mechanisms, gaining access to the systemic circulation [38]. Insulin has been shown to exert mitogenic effects on various cell types involved in wound repair, including fibroblasts and keratinocytes. While these proliferative effects can promote tissue regeneration under normal circumstances, aberrant insulin exposure within the wound microenvironment may lead to dysregulated cellular proliferation, delayed wound closure, and impaired tissue remodeling [39,40,41,42,43,44,45,46]. In veterinary medicine, research on the effects of insulin on wound healing is limited but warrants attention. Studies in animal models have provided valuable insights into the potential risks associated with transcutaneous absorption of insulin in the context of surgical wounds. For instance, a study by Hanson RR et al. [2,22,47] observed impaired wound healing and tissue necrosis in diabetic dogs receiving insulin therapy post-operatively. However, it’s crucial to acknowledge that insulin can also have significant effects on wound healing in non-diabetic individuals, potentially compromising the process and increasing the risk of postoperative complications. This is because insulin can be systemically absorbed, particularly in vascularized tissues undergoing a healing process [38,42,48]. In our study, we did not assess the potential risks associated with the topical application of insulin in the participating patients. This is because none of the patients had diabetes, and none of them had an open wound. Instead, insulin was applied only to the sutured incised tissue. We employed a validated healing scale to meticulously evaluate the progression of surgical wound healing. This scale assessed four key parameters: skin color, hematoma (presence or absence), regional fluid (presence or absence), and wound closure. By objectively assessing these parameters, we gained valuable insights into the dynamics of surgical wound healing [49,50,51,52]. This scale was found to be user-friendly and facilitated the collection of objective data throughout the designated period (8 days), aided by the photographs provided by the patients’ tutors. From all the considered parameters, no significant differences were found at T2 for the skin color, hematoma, and wound closure, contrary to the regional fluid parameter. The change in skin color, hematoma formation, and wound closure time following the topical application of insulin can be attributed to its multifaceted effects on various cellular processes involved in wound healing. One mechanism by which topical insulin may influence skin color is through its vasodilatory effects. Insulin has been shown to enhance blood flow and microvascular perfusion in the skin, which can result in changes in skin coloration [53,54,55]. Additionally, insulin can modulate the activity of melanocytes, the pigment-producing cells in the skin, potentially leading to alterations in skin pigmentation [55]. Regarding hematoma formation, insulin’s anti-inflammatory properties play a crucial role. By suppressing pro-inflammatory cytokines and modulating immune cell function, insulin can attenuate the inflammatory response associated with tissue injury, thereby reducing the extent of hematoma formation. Furthermore, insulin’s promotion of angiogenesis may facilitate the resolution of hematomas by enhancing blood vessel formation and remodeling [56,57]. Although with no statistically significant differences for the wound closure parameter, it was possible to notice that all patients at T2 (8 days post-surgery) presented the wound closure at the Zst contrary to the Zcr. These findings are consistent with previous studies indicating that insulin accelerates the healing process through several mechanisms. Insulin stimulates cell proliferation, collagen synthesis, and epithelialization, leading to faster wound contraction and closure [58,59,60]. Moreover, insulin enhances the production of growth factors such as insulin-like growth factor 1 (IGF-1), which play pivotal roles in tissue regeneration and wound repair [10,61]. We hypothesize that no statistically significant differences were observed for these three parameters at T2 due to the considerable time gap, which may have hindered the accurate assessment of differences. Introducing an intermediate evaluation, such as on day four post-surgery, could potentially enable us to detect variations between these parameters at both regions of the suture, Zcr and Zst. Initially, we considered incorporating an additional time point on day four into the protocol design. However, concerns arose regarding the feasibility of ensuring participant compliance with the additional assessments. Consequently, we opted to limit evaluations to the standard perioperative assessment days (T1 and T8) to ensure consistency and adherence to the project. Respecting the regional fluid parameter, our results are in alignment with the findings of Duckworth et al. [4], as well as with several other studies, including those by Wang, J. & Xu, J.I. [62], Flynn et al. [63], Greenway et al. [64], and Liu et al. [65] which have illustrated that insulin exhibits a remarkable capacity to reduce and mitigate fluid accumulation in wounds, thereby accelerating the overall healing process. This effect is attributed to insulin’s multifaceted actions, including its regulation of inflammatory responses and promotion of tissue regeneration [4] and Bohling. et al. [66], justifying, therefore, the differences registered between both regions at T2. There are other means of wound healing assessment available, including those utilized by Bohling et al. [66]: laser-Doppler perfusion imaging (LDPI), planimetry, direct observation of wounds, and a tensiometer [66]. To evaluate the progression of healing, Chen et al. [39] employed histological observation as an evaluation parameter, along with immunohistochemical and immunofluorescence technology, to monitor the effect of insulin on macrophages in healing [66]. Additionally, in 2017, Dwivedi et al. [67] utilized, among other parameters, the levels of hydroxyproline and hexosamine, constituents of collagen in scar tissue collected from rats [66]. However, because there was no funding for the study, none of these techniques were performed. Additionally, this type of evaluation using these methods would require an invasive procedure in patients for sample collection, which was not justified in the present study design. In our study protocol, an 8 mL syringe with a fast-acting insulin solution in a concentration of 2 IU/mL 0.9% NaCl solution applied with a saturated gauze over the surgical wound study area (Zst) for a period of time of 10 min was used, similar to the technique employed by Paul [68,69]. A lack of consensus among researchers regarding multiple aspects of insulin therapy, such as the type of insulin utilized, dose, and application method, still exists [69,70,71,72,73]. Regarding to the type of insulin, we opted for a standard fast-acting insulin solution, consistent with contemporary approaches described in recent literature. Nevertheless, other studies have employed zinc protamine insulin, which is a slow-acting variant. Respecting the dosage, we used 2 UI/mL for a period of 10 min every 24 h for eight consecutive days. The amount of insulin to be applied to a lesion remains one of the most inconsistent points in the experimental designs of different authors. In 1970, Belfield used concentrations ranging from 10 to 80 IU, with the application period varying from 12 h to 4 days [72]. On the other hand, Paul instituted therapy of 20 IU every 12 h [69]. In 2012, Chen et al. used concentrations of 1.5 IU per milliliter of 0.9% NaCl solution, applying treatment every 24 h [67]. Finally, in 2016, F. Azevedo et al. used doses of 0.5 IU per 100 g of body weight [69]. Considering the application method, we chose to apply the insulin in a solution directed over the wound. In 1970, Belfield [71] utilized an insulin cream directly applied to the lesion, a method also adopted in studies by Lima et al. [70] and F. Azevedo [69]. In 2012, Chen et al. opted for direct application of the insulin solution to the lesion with an absorption period of approximately 5 min [72]. According to F. Azevedo et al. [69], the topical administration of insulin triggers early infiltration of macrophages in the wound area, thereby expediting regional debridement. This macrophage infiltration on the second day is expected to be equivalent to that seen on the third and fourth days in the absence of insulin therapy on surgical wounds [73]. Consequently, the surgical wound enters the proliferative phase of the healing process earlier, typically commencing around the fourth day [73,74]. Subsequently, there is modulation of the inflammatory phase, characterized by an accelerated reduction in the number of macrophages in the region, a process that typically occurs around the fifth day. Ultimately, this leads to earlier re-epithelialization of the surgical wound, ensuring prompt tissue healing and facilitating earlier removal of stitches [26,69,74,75,76]. Insulin shares a sequence very similar to IGF-1 [76]. According to Wicke et al. [75], 90% of wounds lacking IGF-1 exhibit healing disturbances due to alterations in cell replication, collagen deposition, and even the concentration of macrophages in the wound [2]. The effectiveness of insulin in promoting wound healing is attributed to its interaction with insulin-like growth factor 1 (IGF-1) receptors. Both insulin and IGF-1 have long been recognized for their roles in regulating metabolism and growth. They are potent mitogens that promote cell proliferation, migration, and differentiation, thereby facilitating the repair of damaged tissues. IGF-1 is known to play a crucial role in the healing process by modulating various signaling pathways involved in inflammation, angiogenesis, and extracellular matrix remodeling, directly affecting various cell types involved in tissue repair, including fibroblasts, endothelial cells, and keratinocytes [75,76,77]. IGF-1 promotes the proliferation and migration of fibroblasts, which are essential for processes such as collagen synthesis and wound contraction. Additionally, they enhance the proliferation and differentiation of keratinocytes, leading to the formation of new epidermal layers—epithelialization. Furthermore, IGF-1 stimulates angiogenesis (formation of new blood vessels) by inducing the expression of angiogenic factors such as vascular endothelial growth factor (VEGF), thereby promoting the formation of new blood vessels to support tissue repair. Therefore, applying insulin topically to the wound area is believed to stimulate the activation of IGF-1 receptors, thereby enhancing the healing process through the downstream effects on these cells and their functions. In essence, the statement suggests that insulin’s action on wound healing is mediated, at least in part, through its interaction with IGF-1 receptors and the subsequent effects on the cells involved in tissue repair. However, the healing phenomenon is not solely governed by insulin and IGF-1; rather, it involves a complex interplay of multiple factors, including cytokines, chemokines, transcription factors, and endogenous and epigenetic cocatalysts. Understanding the intricate interactions among these factors is essential for elucidating the mechanisms underlying wound healing and developing effective therapeutic strategies. The process is governed by a multitude of factors that interact in a highly orchestrated manner. Cytokines, such as interleukins and tumor necrosis factor-alpha (TNF-alpha), regulate inflammation and immune responses, influencing the recruitment and activation of various immune cells involved in tissue repair. Chemokines guide the migration of inflammatory cells and promote angiogenesis, contributing to the resolution of inflammation and the initiation of tissue remodeling. Transcription factors, such as nuclear factor-kappa B (NF-kappaB) and activator protein-1 (AP-1), regulate the expression of genes involved in inflammation, cell proliferation, and matrix remodeling. Endogenous and epigenetic cocatalysts modulate gene expression and cellular responses to environmental stimuli, influencing the outcome of the healing process. It would have been desirable during the study to quantify the concentration of several of these multiple factors, including cytokines, chemokines, transcription factors, and endogenous and epigenetic cocatalysts, which, in their actions and interactions, are exquisitely variable [39,67]. However, the lack of financial support precluded access to this. Future research should aim to elucidate the underlying mechanisms of insulin’s effects on wound healing and identify potential therapeutic targets for improving clinical outcomes in patients with impaired wound healing.

## 5. Conclusions

In conclusion, wound healing is a complex biological process encompassing distinct stages involving intricate interactions among blood cells, extracellular matrix components, and soluble mediators. Recent advances in wound care, including the use of topical insulin, hold promise for promoting tissue healing. However, this therapy remains relatively unfamiliar to many clinicians in human and veterinary medicine. In our study, we applied a standard fast-acting insulin solution topically with gauze at a concentration of 2 UI/mL for 10 min every 24 h for eight consecutive days. This approach was easily administered by the patient’s tutors. We used a validated healing scale to evaluate the progression of surgical wound healing, focusing on parameters such as skin color, hematoma formation, regional fluid accumulation, and wound closure time. The observed changes in these parameters following topical insulin application highlight insulin’s diverse effects on wound healing processes. We noted significant differences in the healing process evolution of surgical wounds between different regions of the suture, emphasizing the regional variability in response to insulin therapy. These findings align with previous studies demonstrating insulin’s capacity to accelerate wound healing through various mechanisms. Despite these promising findings, further research is needed to optimize the formulation, dosing, and delivery methods of topical insulin for maximal therapeutic efficacy. Additionally, large-scale clinical trials are also needed to validate its safety and effectiveness across different patient populations and wound types. To the best of the authors’ knowledge, this study represents the first investigation into the benefits of topical insulin application in the healing of surgical wounds in cats.

## Figures and Tables

**Figure 1 animals-14-01358-f001:**
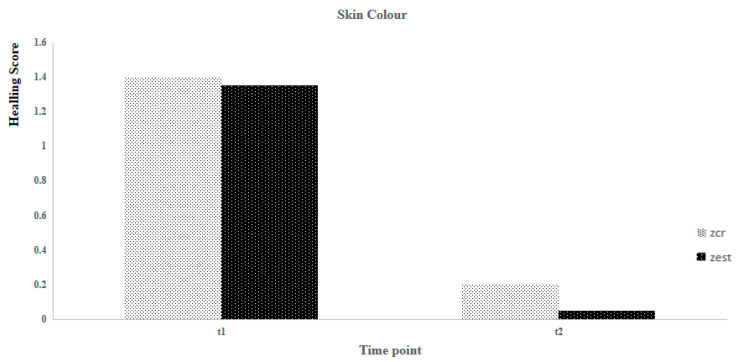
Comparison between Zst and Zcr of wound healing evolution considering the skin color parameter.

**Figure 2 animals-14-01358-f002:**
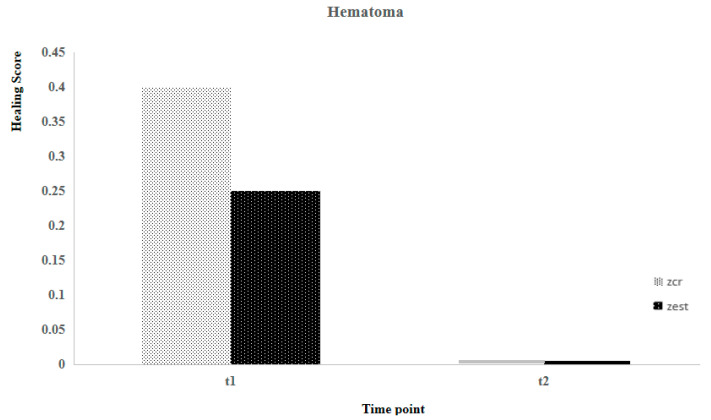
Comparison between Zst and Zcr of wound healing evolution considering the regional fluid parameter.

**Figure 3 animals-14-01358-f003:**
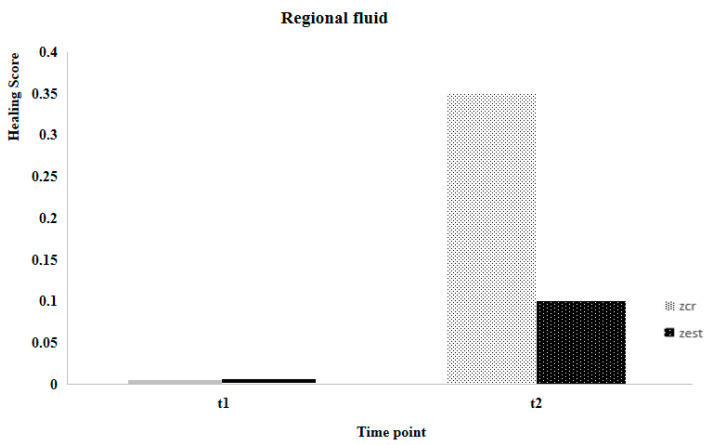
Comparison between Zst and Zcr of wound healing evolution considering the presence or absence of hematoma.

**Figure 4 animals-14-01358-f004:**
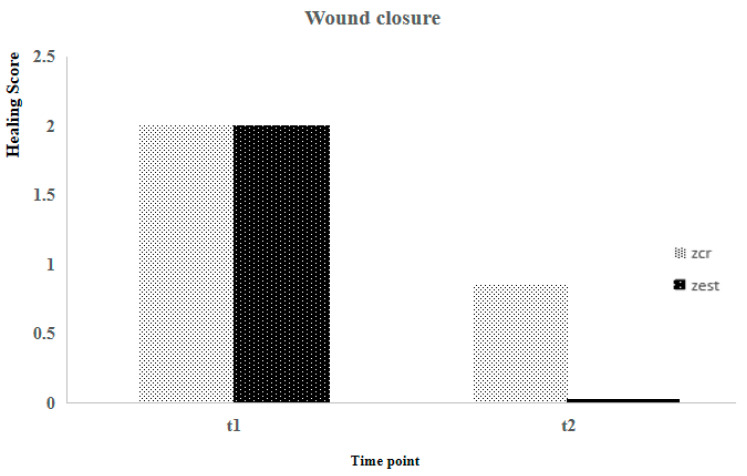
Comparison between Zst and Zcr of wound healing evolution considering the wound healing closure.

**Figure 5 animals-14-01358-f005:**
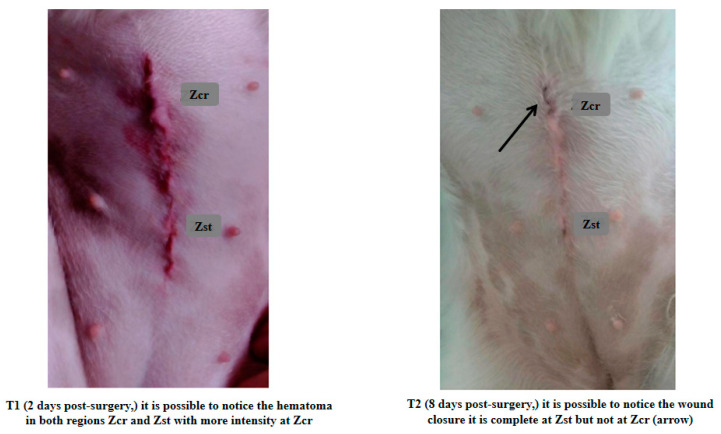
Comparision between Zst and Zcr of wound healing evolution considering the both time-points of the study (T1–2 days post-surgery and T2–8 days post-surgery). It is possible to notice that at T2 the Zst presents a total wound healing closure, but not at Zcr (arrow).

**Table 1 animals-14-01358-t001:** Post-surgical healing evolution scale parameters: (1) skin color, (2) hematoma (presence or absence), (3) regional fluid, and (4) wound closure. Each of the parameters is rated on an ordinal scale, and the final score is the result of the sum of each considered parameter. The higher the injury or change level that tissue presents, the higher the final scale score.

Post-Surgical Healing Evolution Scale	
Parameter	Rate
Skin color	0–4
Presence or absence of hematoma	0–1
Presence of regional fluid	0–4
Wound closure	0–2
Final score	0–11

**Table 2 animals-14-01358-t002:** Sample and Shapiro-Wilk normality test for sample parameters.

Parameter		Median	IQ	Shapiro-Wilk Test
				*p*-Value
Age (months)		9.0	2.57	0.015 *
Body-weight (kg)		4.03	0.24	0.920
Wound Parameter Evaluated with the Post-Surgical Healing Evolution Scale	Time Point	Median	IQ	Shapiro-Wilk test
Skin color	T1	1.37	0.40	0.939
	T2	0.12	0.33	0.939
Hematoma	T1	0.30	0.46	0.939
	T2	0.00	0.00	0.939
Regional fluid	T1	0.00	0.00	0.939
	T2	0.22	0.42	0.939
Wound closure	T1	0.07	0.26	0.939
	T2	0.15	0.42	0.939
Final Scale Score	T1	1.74	0.86	0.939
	T2	0.50	0.71	0.939

* Statistically significant.; Interquartile (IQ).

**Table 3 animals-14-01358-t003:** Kruss-Wallis test and Tukey HSD test to evalute statistically significant differences for healing scale final score between the Zcr and Zst, and between each parameters they were registered.

Parameter	Kruss-Wallis Test		Tukey HSD		
	H-Statistic	*p*-Value	Parameter	Q-Statistic	*p*-Value
T1 final score for Zst and ZctZcr	0.123	0.725	T1 Skin color at Zst vs. Zcr	−1.407	0.261
			T1 Hematoma at Zst vs. Zcr	3.500	0.071
			T1 Regional fluid at Zst vs. Zcr	3.466	0.104
			T1 Wound closure at Zst vs. Zcr	2.258	0.638
T2 final score for Zst and Zcr	5.24	0.022 *	T2 Skin color at Zst vs. Zcr	2.029	0.159
			T2 Hematoma at Zst vs. Zcr	1.414	0.323
			T2 Regional fluid at Zst vs. Zcr	3.526	0.017 *
			T2 Wound closure at Zst vs. Zcr	1.042	0.465

* Statistically significant.

## Data Availability

The original contributions presented in the study are included in the article/supplementary material, further inquiries can be directed to the corresponding author.

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
