# Peer review of "The Use of Fast-Acting Insulin Topical Solution on Skin to Promote Surgical Wound Healing in Cats"

_animals, 2024, doi:10.3390/ani14091358_

Round 1
Reviewer 1 Report
Comments and Suggestions for Authors
Due to multiple points of homology across mammalian species, the subject matter explored in this Manuscript is undoubtedly of high interest to those involved in the surgical wound healing of animal companions as well as their owners. In drilling down to the molecular landscape, Authors astutely touch upon a relevant feature by noting "IGF-1, which is only expressed in fibroblast and keratinocyte cells of the dermis and epidermis. IGF-1, a protein with a sequence very similar to insulin, has been shown to promote granulation tissue synthesis in in vivo studies."
Likewise, there is salient thought in Authors further observation that "....[I]n 2012, Abo El Asrar et al. demonstrated a relationship between IGF-1 levels and inflammation markers, namely IL-6 and IL-8, in diabetic patients, highlighting decreased IGF-1 levels and increased inflammation markers in such patients."
It is at this point that Reviewer’s concerns arise. For example, transcutaneous absorption of insulin may indeed represent a relatively minor risk. However, in a fresh surgical wound, particularly one placed in such a core anatomical site, have Authors considered and controlled for the risk of toxicity?
Next, and apparently without reference to cite, Authors confidently state “Recently, the topical application of growth factors as promoters of healing has become quite common. However, these factors often exist in low concentrations in the wound bed due to decreased supply or natural degradation.” As numerous cites tend to contradict the assertion, it would be best to rephrase or redact.
From here, Reviewer finds several significantly troubling aspects to the Authors’ overall methodology. For example, Authors report that “…[a]ccording to F. Azevedo et al., the topical administration of insulin triggers early infiltration of macrophages in the wound area, thereby expediting regional debridement.” Indeed, Azevdeo et al appear to have undertaken a number of scientifically rigorous analytic steps in their work — yet none of these appear to have been reproduced in the work described in the submitted Manuscript. Their less than scientifically rigorous approach is perhaps best evinced by noting that at time point T2 therapy was apparently dispensed, not by Author investigators, but rather by the pet cat owners.
Further evidence of a less than strenuous approach also arises in Authors’ leap of logic where they assert that “…[t]he significance of IGF-1 in healing stems from its direct effects on fibroblasts, endothelial cells, and keratinocytes.” All well and good on its face. However, the problem occurs in Authors’ next sentence “…Consequently, the topical application of insulin to the wound area ensures its action through its relationship with IGF-1 receptors in the injured region.”
Here, Authors inadvertently stumble into a major category error. Indeed, IGF-1 constitutes a well described healing factor. However, to focus primarily on this marker is to ignore the complex interplay of multiple factors, including cytokines, chemokines, transcription factors, endogenous and epigenetic cocatalysts which, in their actions and interactions are exquisitely variable. The confluence of poor planning, methodological failures, and leaps of logic render the entire work irredeemably tainted.
Author Response
Dear Reviewer,
I hope you are well. I'm pleased to inform you that the manuscript titled "THE USE OF FAST-ACTING INSULIN TOPICAL SOLUTION ON SKIN TO PROMOTE SURGICAL WOUND HEALING IN CATS" submitted to Animals has been revised and corrected based on your feedback. I have attached the revised manuscript along with responses to each comments.
Best regards,

Reviewer 2 Report
Comments and Suggestions for Authors
This is an informative research article revealing the effect of Topical insulin on surgical wound healing specific in cats.
There are some suggestions for authors.
(1) Line 47~48 "However, ...... or natural degradation"
There are some dressing system that may overcome this problem.
If there are limiting factors for growth hormone, why authors still considered insulin as a good candidate for promoting wound healing? Can insulin overcome the limiting factors described in the sentence?
(2) For the introduction section, It would be better to give a introduction for the use of topical insulin in other animal models like rats and human clinical practice.
(3) line 90~93 "For the evaluation ...... to the suture stitches"
Have author consider wound closure time as another indicator to evaluate the effect of insulin in wound healing in this study?
(4) Figure 4, please indicate that whether there is a significant difference at T1 for the factor "Presence or Absence of Hematoma" between "Zst" and "Zct". If yes, please include a discussion about why they can finally reach the same end point.
(5) For Discussion, could author also include your thoughts about the reasons why not significant differences were found in other three factors (Skin color, hematoma, suture stitches). Is it possible you may find differences at the time points earlier than T8?
Author Response

(The authors gave the same response as above.)

Reviewer 3 Report
Comments and Suggestions for Authors
Please find attached the revised document.

Author Response

(The authors gave the same response as above.)

Round 2
Reviewer 3 Report
Comments and Suggestions for Authors
Author Response
Dear Reviewer,
I hope you are well. I'm pleased to inform you that the manuscript titled "THE USE OF FAST-ACTING INSULIN TOPICAL SOLUTION ON SKIN TO PROMOTE SURGICAL WOUND HEALING IN CATS" submitted to Animals has been revised and corrected based on your feedback. I have attached the revised manuscript along with responses to each comments.
Best regards,
Abstract Lines 36-37: The authors pointed out that significant differences in the healing process were observed between the control and study zones, highlighting the regional variability in response to insulin therapy. Be more precise, focus on the results obtained.
We completely agree with the reviewer's emphasis on the importance of providing more precise information about the results obtained from the application of insulin by tutors. Consequently, we have decided to incorporate the following sentence into the Abstract section:"Statistically significant differences were registered in the final healing scale scores between Zcr and Zst (p < 0.022), as well as for the parameter of regional fluid (p-value = 0.017). Additionally, at T2, all Zst regions exhibited wound closure, whereas Zcr did not, although not in a statistically significant manner. The observed discrepancy at T2 between the Zcr and Zst regions may suggest a potential benefit of utilizing insulin. No side effects resulting from the insulin topical application performed by the tutors were recorded in the Zst suture group.”
Lines 38-42: It is redundant for this section Please delete.
Corrected
Figures Please put a thicker black and/or gray line on the x-axis for each group with a score of 0 in the Figures, because this is how the Figures appear to be incomplete.
Corrected
Figure legend: replace “zest” with “zst”
Corrected
